

# Attenuation of cosmic-ray up-scattered dark matter

**Helena Kolesova⋆**

University of Stavanger

⋆ helena.kolesova@uis.no

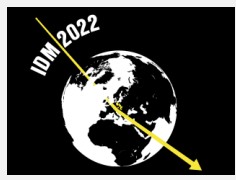

## Abstract

**GeV-scale dark matter particles with strong coupling to baryons evade the standard direct detection limits as they are efficiently stopped in the overburden and, consequently, are not able to reach the underground detectors. On the other hand, it has been shown that it is possible to probe this parameter space taking into account the flux of dark matter particles boosted by interactions with cosmic rays. We revisit these bounds paying particular attention to interactions of the relativistic dark matter particles in the Earth's crust. The effects of nuclear form factors, inelastic scattering and extra dependence of the cross section on transferred momentum (e.g. due to presence of light mediators) are studied and are found to be crucial for answering the question as to whether the window for GeV-scale strongly interacting dark matter is closed or not.**

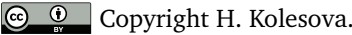



## 1 Introduction

Direct detection experiments are trying to shed light on the nature of dark matter (DM) but, although great progress was made in past years [1–3], their reach in the space of DM mass and couplings is still limited. In particular, DM interactions with nucleons are probed by experiments looking for collisions of DM with nuclei in the detector which requires a certain minimal DM kinetic energy to trigger a detectable signal. Given the fact that halo DM particles reach the Earth at velocities of the order of $\sim 10^{-3} c$, such particles don't attain sufficient kinetic energy if their mass is too low. Hence, sub-GeV DM is typically not probed by standard direct detection experiments. Another "blind spot" for standard direct detection experiments is a result of the fact that if DM interacts too strongly with nuclei, it is efficiently stopped in the Earth's crust and does not reach the underground detectors. This latter issue was addressed, e.g. by the dedicated CRESST surface run [4] in the context of direct detection. Additionally, strongly coupled sub-GeV DM has been further constrained by the possible effects on structure

formation [5, 6] or on the cooling of gas clouds near the Galactic Centre [7]. Nonetheless, state of the art probes may still leave room for strongly interacting DM candidates like the stable "sexaquark" state with a mass around 2 GeV [8].

In this work we concentrate on yet another constraint on DM with strong couplings to baryons. Namely, it was shown in Ref. [9] that collisions of cosmic ray (CR) nuclei with such DM in the Galactic halo result in a flux of relativistic DM particles coming to Earth (CRDM flux). These particles can trigger detectable signal in standard direct detection experiments despite their sub-GeV mass. In this way, the Xenon-1T limits [3] were reinterpreted to constrain the spin-independent DM-nucleon scattering cross sections roughly between $10^{-31}$ and $10^{-28}$ cm$^2$ for DM masses up to about 2 GeV [9]. It is worth stressing that also in the CRDM case, the upper boundary of the excluded region is set by the fact that DM coupled too strongly to nucleons cannot reach the underground detectors. In this text, we focus on the attenuation of the CRDM flux in the Earth's crust, and we show that a more precise treatment (described in section 2) leads to extension of CRDM limits to larger DM masses. The consequence of this is to close the parameter space for DM-nucleon cross sections exceeding $10^{-30}$ cm$^2$ (see section 3). As discussed in section 4, we checked that our conclusions hold for a range of generic DM scenarios such as those where interactions with nucleons proceed via light mediators. While the main results are highlighted in this text, the technical details of the modeling and particle physics scenarios can be found in [10]. For the analysis performed in this work we used the numerical tool DarkSUSY [23] and the updated routines will be included in the next public release of this code.

## 2 Attenuation of the CRDM flux in the Earth's crust

The evolution of the DM kinetic energy $T_\chi^z$ at depth $z$ can be described by the energy loss equation:

$$\frac{dT_\chi^z}{dz} = -\sum_N n_N \int_0^{\omega_\chi^{\max}} \mathrm{d}\omega_\chi \frac{\mathrm{d}\sigma_{\chi N}}{\mathrm{d}\omega_\chi} \omega_\chi \,, \tag{1}$$

where the sum runs over the nuclei $N$ in the overburden, each with a number density $n_N$ and a differential cross section $\mathrm{d}\sigma_{\chi N}/\mathrm{d}\omega_\chi$ describing the scattering with DM particles in terms of the kinetic energy lost by the DM particle, $\omega_\chi$. It is this cross section that has to be treated more precisely in order to obtain realistic predictions as to the parameter space that is excluded by the non-observation of the CRDM component in detectors like Xenon-1T. In particular, we concentrate on detailed modeling of the nuclear form factors in the elastic contribution to $\mathrm{d}\sigma_{\chi N}/\mathrm{d}\omega_\chi$ and on the effect of including inelastic scattering in following sections 2.1 and 2.2, respectively.

### 2.1 Effect of nuclear form factors

For the calculation of the elastic contribution to the DM-nucleus scattering cross section, we follow the approach of standard direct detection experiments that translate their observations into limits on the spin-independent DM-nucleon cross section $\sigma_{SI}$ using its following relation to the differential DM-nucleus cross section:

$$\frac{\mathrm{d}\sigma_{\chi N}}{\mathrm{d}\omega_\chi}\bigg|_{\mathrm{el}} = A^2 \frac{\mu_{\chi N}^2}{\mu_{\chi P}^2} \times \frac{\sigma_{\mathrm{SI}}}{\omega_\chi^{\max}} \times G^2(Q^2) \,. \tag{2}$$

Here $\mu$ refers to reduced mass of the given 2-particle system, $\omega_\chi^{\max}$ is the maximum energy lost by the DM particle[1] and $A$ is the atomic mass number of the nucleus. The factor $A^2$ then captures the coherent enhancement of the scattering cross section characteristic for spin-independent couplings of DM to nucleons (under the assumption of an equal coupling of DM to protons and neutrons). Nuclear form factor $G(Q^2)$, on the other hand, expresses the loss of coherence across the nucleus for large momentum transfers $Q^2 = 2m_N\omega_\chi$. Since, especially for heavy nuclei, $G$ is a steeply falling function of $Q^2$, the form factors lead to a significant reduction of the elastic cross section in the case of relativistic CRDM particles, i.e. reduced attenuation of the CRDM flux. The effect of nuclear form factors in the attenuation part was neglected in the initial study [9], but was added in a later re-analysis [11]. In our work we identify the importance of the form factors for setting CRDM limits, and compared to [11], include the more accurate model-independent form factors [12]. On the other hand, we point out below that the almost vanishing cross section at large momentum transfers, as obtained when considering only the contribution of Eq. (2), is unphysical since the additional contribution of inelastic scattering becomes relevant for CRDM particles scattering on nuclei.

## 2.2 Effect of inelastic scattering

Although the CRDM flux peaks for kinetic energies between 10 and 100 MeV depending on DM mass (see [10] for details), a significant amount of DM particles with kinetic energies larger than 100 MeV may arrive on Earth. For these, momentum transfer can be large enough to resolve individual nucleons or even partons in the scattering process and the contribution of inelastic scattering may easily dominate the right-hand side of Eq. (1). Indeed, by comparison with analogous processes in case of neutrino-nucleus scattering at comparable momentum transfer (see, e.g. [13] for a review), DM particles with kinetic energies $T_\chi \gtrsim 0.1$ GeV are expected to effectively scatter off individual nucleons for large energy transfers $\omega_\chi$ (via so-called quasi-elastic scattering), for $T_\chi \gtrsim 0.3$ GeV the excitation of hadronic resonances becomes possible and, finally, for kinetic energies of a few GeV, deep inelastic scattering becomes the relevant contribution at large $\omega_\chi$. Calculation of the corresponding cross sections has to take into account the effect of the nuclear environment (like the nuclear potential or spin statistics) and cannot, hence, be easily performed analytically. For this reason, we estimate the inelastic contribution to $d\sigma_{\chi N}/d\omega_\chi$ by first using the numerical code GiBUU [14] to calculate neutrino-nucleus cross sections. We then rescale the result appropriately in order to take into account the properties of DM scattering [10]. Although this procedure introduces additional uncertainty in the scattering cross section, we checked that irrespective of the precise implementation, the inelastic scattering leads to a significant increase of the stopping power in soil compared to what was assumed in [11] and, hence, a significant reduction in the size of the excluded region, as described below.

## 3 Exclusion limits

In the section, we present the results for the case of a "constant" DM-nucleus cross section (2).[2] As can be seen in Fig. 1, CRDM limits including both the effect of form factors and inelastic scattering in the attenuation part are stronger than the conservative limits of [9], especially

---

[1]In the case of elastic scattering, $\omega_\chi$ is equal to the kinetic energy of the recoiled nucleus $T_N$ and for a given nucleus mass, $\omega_\chi^{\max} = T_N^{\max}$ is uniquely determined by the DM kinetic energy and mass.

[2]The differential cross section where the only dependence on the transferred momentum comes from the nuclear form factors is a good approximation for contact interactions in the highly non-relativistic limit. For the CRDM particles, full $Q^2$-dependence of the cross section may influence the final results, see the comments in section 4.

for heavier DM where the form factors play significant role. On the other hand, the results of [11] clearly overestimate the excluded region. One can see that the true limits correspond roughly to the case where inelastic scattering is neglected, but at the same time, the CRDM flux is artificially cut at kinetic energies around 0.2 GeV. This confirms our findings related to the attenuation of the CRDM flux, namely that DM particles with $\mathcal{O}(0.1)$ GeV kinetic energies become efficiently stopped by inelastic scattering.

In the right panel of Fig. 1 we compare the CRDM excluded region to limits based on the Lyman-$\alpha$ forest [5], the Milky Way satellite population [6], gas clouds in the Galactic Centre region [7], the XQC experiment [15, 16], and a recently analysed storage dewar experiment [17, 18]. We also present the limits based on the CRESST surface run [4, 19] (solid green lines), together with the alternative limits based on the assumption of a thermalization efficiency of $\epsilon_{\mathrm{th}} = 2\,\%$ [16] and $\epsilon_{\mathrm{th}} = 1\,\%$ [20] (green dashed and dash-dotted lines, respectively), which is significantly more pessimistic than the one adopted in the CRESST analysis.

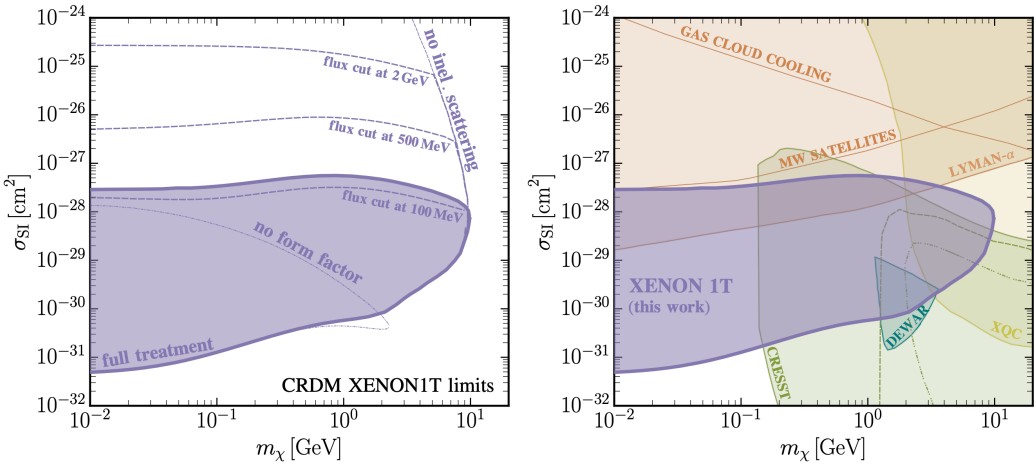

Figure 1: Limits on a constant spin-independent DM-nucleon scattering cross section as a function of the DM mass (solid lines). In the left panel, dash-dotted lines show the excluded region that results when assuming a constant cross section in the attenuation part (as in Ref. [9]). Dashed lines show the effects of adding form factors in the attenuation part, but no inelastic scattering, resulting in limits similar to those derived in Ref. [11]. For the latter case, for comparison, we also show the effect of artificially cutting the incoming CRDM flux at the indicated energies. In the right panel, we compare the CRDM limits to other published constraints.

## 4  Discussion and conclusions

As can be seen in Fig. 1, irrespective of the thermalization efficiency assumed for the CRESST experiment, there is no parameter space left unconstrained for DM-nucleon cross sections exceeding $10^{-30}$ cm$^2$ in the entire MeV to GeV DM mass range.[3] Of course, the simplified DM-nucleus cross section assumed here is in contrast to the more complex dependence of the cross section on the transferred momentum and invariant mass that is expected for realistic sub-GeV DM models. Several of these more generic DM scenarios were considered in our study [10], namely, DM interacting via scalar or vector mediators with MeV-to-GeV masses and finite-size DM. The extra $Q^2$-suppression of the cross section leads to a decrease in the CRDM flux in

---

[3]Note that strongly coupled DM lighter than about 10 MeV is nonetheless excluded by the BBN constraints [21].

certain cases, consequently, a tiny open parameter space can appear for a narrow range of mediator masses, but only if the CRESST thermalization efficiency was indeed as low as 2%. We note that such an assumption is not supported by data or simulations [22]. Our conclusion is, hence, that there is generically no room to hide for sub-GeV DM with DM-nucleon cross sections exceeding $10^{-30}$ cm$^2$.

# Acknowledgements

HK thanks James Alvey and Torsten Bringmann for a fruitful collaboration and the organizers of the IDM conference for the opportunity to present this work.

**Funding information** HK is supported by the ToppForsk-UiS Grant No. PR-10614.

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
