# Peer review of "Attenuation of Cosmic-Ray Up-Scattered Dark Matter"

_SciPost Physics Proceedings, doi:SciPost Phys. Proc. 12, 055 (2023)_

## Round 1 · Referee Report · Anonymous (Referee 1) · 2022-10-20

Strengths

  1. The topic of this work, cosmic ray up-scattering, has been suggested as a very useful mechanism to boost astrophysical dark matter that strongly couples with e/N, leading to detectable recoil events in large-volume detectors.

  2. This manuscript carefully takes into account the effects of inelastic scattering with the Earth, among other effects, significantly improving the precision of the relevant DM direct detection bounds.

Weaknesses

The crucial point is the inclusion of inelastic scattering with the Earth, which seems to rely on the results of neutrino-nucleus cross sections. But neutrino is supposed to always relativistic, while here dark matter is only semi-relativistic, so it would be nice if the author can provide more comments on the relevant uncertainties here.

Report

This manuscript summaries their very useful improvements on the treatment of semi-relativistic dark matter scattering with the Earth, by including several subtle effects. It is quite clear, and meets the criteria of SciPost Physics Proceedings.

And it would be even better if the author can further improve it by making the requested changes below.

Requested changes

Although technical details should not be required for a proceeding, given the importance here, I suggest the author add a few sentences in Sec. 2.2, to comment on the uncertainties in their treatment on inelastic scattering.

PS, the author may shorten the first paragraph of Introduction a little, if the manuscript becomes too long after the changes.

  • validity: high
  • significance: high
  • originality: top
  • clarity: high
  • formatting: perfect
  • grammar: excellent

Author:  Helena Kolesova  on 2022-11-15  [id 3018]

(in reply to Report 1 on 2022-10-20)
Category:
answer to question

Hereby, I would like to thank the referee for useful remarks. I changed the text accordingly (see my recent resubmission), however, I would also like to give here a self-contained answer to the referee's question.

The relevance of different inelastic processes for scattering with nuclei is mainly determined by the momentum transferred in the interaction. Maximum transferred momentum, in turn, is given by the kinetic energy of the scattering particle. Although the DM particles under consideration may only be semi-relativistic, their kinetic energies are comparable to kinetic energies for neutrinos that scatter inelastically from nuclei. Consequently, we believe that qualitatively, similar inelastic processes can be expected for DM. Of course, the exact value of the DM-nucleus inelastic cross section depends on the nature of the DM-nucleon interactions which may differ from the neutrino ones. We tried to capture this by rescaling the cross section by the ratio of DM-nucleon and neutrino-nucleon cross sections, but of course, a certain level of uncertainty is introduced in this way. We checked, however, that this uncertainty does not have a large impact on our final conclusions.

---

## Round 2 · Author Response

Based on useful comments of the referee, I made several modifications in Section 2.2.

---

## Round 2 · List of Changes

• I clarified that the appearance of different inelastic processes depends on momentum transferred in the scattering.
  • I commented on the fact that the uncertainties in modeling the inelastic cross section don't affect the main conclusions of this paper.

---

## Editorial Decision

published